# Healthcare-seeking behaviours among mother's having under-five children with severe wasting in Dodoma and Mbeya regions of Tanzania-A qualitative study

**Elizabeth J. Lyimo[1], Maria Msangi[1], Anna J. Zangira[1], Rose V. Msaki[1], Aika Lekey[1], Magreth Rwenyagira[1], Ramadhan Mwiru[2], Patrick Codjia[2], Mangi Ezekiel[3], Germana H. Leyna[1,3], Ray M. Masumo[1] ***

**1** Tanzania Food and Nutrition Centre, Dar es Salaam, Tanzania, **2** The United Nations Children's Fund (UNICEF) Tanzania, Dar es Salaam, Tanzania, **3** Muhimbili University of Health and Allied Sciences (MUHAS), Dar es Salaam, Tanzania

* rmasumo@yahoo.com

## Abstract

Maternal healthcare-seeking behaviour affects the health and well-being of under-five children. Drawing from the concepts of the health belief model, this study seeks to understand the determinants of health-seeking behaviours among mothers or caregivers of under-five-year-old children having severe wasting in Tanzania. A qualitative study employing the ethnography method conducted 32 semi-structured and narrative interviews with healthcare workers, community health workers, traditional healers, religious and village leaders, and mothers or caregivers of children who had acute malnutrition. The analysis of transcripts was done by qualitative content analysis. Further, the thematic analysis was carried out by assigning data into relevant codes to generate categories based on study objectives. Severe wasting among under-five-year-old children was not observed as a serious disease by the majority of mothers or caregivers. The study established that the health systems parameters such as the availability of the community health workers or healthcare providers and the availability of medicines and supplies to the health facility impact on mothers' or caregivers' healthcare-seeking behaviours. The findings also show that long distances to the health facility, behavioural parameters such as lack of awareness, negative perception of the management of severe wasting at the health facility, superstitious beliefs, women's workload, household food insecurity, and gender issues have a significant role in seeking healthcare. The results reaffirm how a programme on integrated management of severe wasting in Tanzania should encompass sociocultural factors that negatively influence mothers or caregivers of children with acute malnutrition. The programme should focus on engaging community structures including traditional healers, religious and village leaders to address prevailing local beliefs and sociocultural factors.

**Data Availability Statement:** All datasets underlying this study are freely available at the public repository https://osf.io/sk735.

**Funding:** The authors received no specific funding for this work.

**Competing interests:** The authors have declared that no competing interests exist.

## Introduction

It is well established that children with severe wasting are a major global public health concern and have a higher risk of death from common childhood illnesses such as diarrhoea and malaria [1]. The global average under-five years mortality rates has been reduced from 93 per 1,000 live births in 1990 to 39 per 1,000 live births in 2018 [2]. However, about three-quarters of the under-five years' child deaths occurred in Sub-Saharan Africa, as well as Central and Southern Asia, accounting for only half of the under-five year's population globally [2]. Nutrition-related factors contribute to about 45 percent of the under-five years child deaths globally and, Tanzania is the home to more than 500,000 under-five-year-old children who are suffering from severe wasting, accounts 3.5 percent of all Tanzanian children of age between 0–59 months [1, 3].

The World Health Organization (WHO) recommends the use of a cut-off for weight-for-height of below -3 standard deviations (SD) of the WHO growth standards, or a mid-upper arm circumference < 115 mm, or by the presence of nutritional oedema to identify under-five-year-old children as having severe wasting [4]. Children with severe wasting have weakened immunity, susceptible to long-term developmental delays, and face an increased risk of death from a common childhood illness and, thus urgent therapeutic feeding, treatment, and care for survival are crucial [1].

Over the past 10 years, Tanzania has been implementing an integrated management of severe wasting (IMAM) that links inpatient treatment of malnourished children (those with complications) with outpatient care (severe without complications) and comprehensive community involvement and mobilization to facilitate access to health services and to enhance knowledge transmission in the community [3]. The IMAM programmes focused on disseminating biomedical knowledge about severe wasting and, mostly not considering or integrating mothers' or caregivers' customs and beliefs, which are widespread in Tanzanian communities [3, 5–8]. There is a body of knowledge on various behavioural change theories that effectively describe dimensions of human behaviours with respect to health seeking [5–9]. However, there are few studies that explain the hierarchy within the behavioural determinants with regard to the manner in which they play the role to influence behavioural outcomes [9].

Research on health-seeking behaviours (HSB) in LMICs suggested that inappropriate HSBs are skewed among different population segments. Almost 70 percent of pregnant women in Kenya belonging to the upper socio-economic stratum were found to have their deliveries in health facilities, compared with 42 percent among pregnant women in the middle socio-economic stratum and 38 percent in the low socio-economic stratum [10]. A study across six African countries which are Democratic Republic of Congo, Ethiopia, Malawi, Niger, Sudan and South Sudan reported that the common determinants of seeking Community Base Management Acute Malnutrition (CMAM): Fear of rejection, lack of identity in the child's condition as malnutrition, lack of confidence in the programme, relapse and distance to the healthcare facility. Similar findings were observed from the evaluation of severe wasting treatment done in the twenty-one countries that, lack of knowledge of malnutrition, lack of knowledge about CMAM, high opportunity costs and distance to healthcare facilities were barriers to care seeking [11, 12]. The literature available clearly establishes that determinants of health-seeking behaviours can be broadly classified into two groups. The first group is composed of studies that emphasise the utilisation of the formal system. The studies that fall under this category involve the development of models that describe the series of steps people take toward healthcare seeking. These models are sometimes referred to as 'pathway models' [13]. While there are several variations of these models, the Health Belief Model and Andersen's Health Behaviour Model are often used as a basis in discussions involving HSB [13]. The second group

comprises those studies which emphasize the process of illness response, or health-seeking behaviour. These studies demonstrate that the decision to engage with a particular medical channel is influenced by a variety of factors such as socioeconomic status, sex, age, social status, types of illness, access to services, and perceived quality of the services [14]. A majority of the studies under this second category focus on specific genres of determinants that lie between patients and services such as geographical, social, economic, cultural and organizational factors [13].

In Tanzania, the determinants of health-seeking behaviours have been investigated in relation to other disease conditions [15, 16]. Though research on the determinants of health-seeking behaviours among caretakers with malnourished children has been investigated elsewhere in Africa, there is still some scope for understanding better the determinants of health-seeking behaviours [17]. Therefore, this study seeks to understand the determinants of health-seeking behaviours among mothers or caregivers of under-five years' old children having severe wasting in Tanzania. Thus, the findings of this study will enhance the understanding of the determinants that impact healthcare-seeking behaviours, the roles of each community member and make recommendations to the policymakers, researchers, implementers and other stakeholders. The findings will also serve the purpose of bringing in a conceptual understanding of the healthcare-seeking behaviours related to severe wasting. That would significantly help policymakers in designing better healthcare solutions and more effective communications.

## Methods

### Ethics statement

This study obtained ethical permit from the National Health Research Ethics Review Committee (NatHREC) at National Institute for Medical Research (NIMR) with reference no. NIMR/HQ/r.8a/Vol.IX/3964. We also obtained permission to conduct a study at regional, district and ward levels in both regions. All study participants provided written consent prior to participating in the study and, participants did not receive financial compensation for participating. All data were anonymised and, will be destroyed when the project is concluded.

### Study design

A qualitative study employing the ethnography method was conducted to explore information from Key informant interviews (KIIs) and Focus Group Discussions (FGDs), and semi-structured interview guiding questions were used to capture the views and understandings of community members and mothers/caregivers of children less than five years with severe wasting.

### Study setting

The study was purposively conducted in Dodoma region (Chemba and Chamwino districts), and Mbeya region (Kyela and Mbeya districts). The regions and districts were selected based on case load using purposive sampling guided by the recommendations of the simulation and guidelines for sample sizes in qualitative research such as (1) Identifying a population of information sources and sub-populations; (2) Estimating an order of magnitude of the number of codes per sub-population; (3). Estimating the mean probability of a code being observed and; (4) Assessing which scenario is most applicable to each sub-population [18]. We conducted the study in eight villages purposefully selected in four wards of Kyela DC, Mbeya DC, Chamwino DC, and Chemba DC, which has been conducted in other similar settings [17]. Participant's recruitment was carried out from May 8th 2022 to June 17th 2022.

## Selection of study participants

Mothers/caregivers of under-five-year-old children who had severe wasting were identified by community health workers (CHWs) from their records. These were leaders who knew the area and could identify households with young children who had malnutrition within six months. Mothers/Caregivers currently receiving outpatient treatment for severe wasting in Reproductive and Child Health (RCH) were accessed in their health facilities in the respective study area/council [18, 19].

We also recruited a group of influential personalities in the villages (village and religious leaders, traditional healers), Community Based Organisation (CBOs) and healthcare workers who have influence over mothers/caregivers. They were purposively identified by district nutritionist and local leaders who are known to the study team and respondents in the FGDs. The number of KIIs and FGDs was determined by data saturation [18, 19].

## Data collection

In each ward, two FGDs and six KIIs were conducted.

**Focus Group Discussions (FGDs).**   These were groups of mothers/caregivers of children below five years who have or had acute malnutrition. In the FGDs, the total number of participants ranged from 6 to 12. Community leaders assisted with identification of the FGD participants in their respective communities/villages. The FGDs interview guide is annexed as *S1 Appendix*.

**Key informant interviews (KIIs).**   KIIs were conducted with mothers/caregivers of children who had severe wasting; CHWs and the influential personalities (religious leaders, traditional healers and local government/village leaders), CBOs, healthcare providers and mothers/caregivers in the respective communities. The purposive sampling techniques were used to select mothers/caregivers of malnourished children for KII. Village registers were used to identify mothers/caretakers of malnourished children with the help of the CHWs in the respective village area. The interviewing process continued until a point was reached whereby no new information emerged- data saturation point. The KII interview guide is annexed as *S2 Appendix*.

The FGDs and KII were designed to capture information on awareness, perceptions and factors contributing to poor adherence to the management of severe wasting. Prior to study, teams were oriented on the tools and rehearsals were done in a nearby community to become more familiarised with the questions and identify areas for improvement [19–21].

The FGDs and KIIs were administered in a quiet place with enough privacy to allow respondents to speak freely and assure that the information provided was confidential and only accessible to the research team. The FGDs among parents also included men (in at least two sessions). Written informed consent was sought from each study participant at the start of each FGD and, participant who refused to sign a written consent were excluded from the study. In addition, we asked for permission to use audio recorders to record the conversations. Each group had one moderator and one note-taker that were trained in aspects of qualitative research and participated in pre-test prior to data collection. Notebooks were used to assist in capturing non-verbal communication and the other relevant context of the discussion/interviews, which had been recognized in other similarly studies [19–21].

## Data analysis

Verbatim transcription of the FGD and KII sessions was done by research assistants or note-takers. The qualitative personnel verified all transcriptions to ensure that transcription was accurate by listening to the audio and make sure what was written were similar to what was in

the audio [19]. The interviews were conducted in the Swahili language and simultaneous transcription and translation were used [22].

The interviews after being transcribed verbatim were coded thematically for content analysis and interpretation. The team of researchers under the leadership of persons with expertise in qualitative methods read the text several times before developing codes [19, 22]. The second person (also trained and experienced) coded a subset of transcripts using the developed codebook in an attempt to assess the quality of the data and their reliability for interpretation and final use [22]. The codebook structure was developed iteratively after the completion of transcription. The codebook is annexed as *S3 Appendix*. Themes were discussed and agreed upon by all researchers. The thematic analysis was carried out by assigning data into relevant codes to generate categories based on study objectives [19].

## Results

Various themes in the result section were generated from the transcripts.

### Characteristics of study participants

As depicted in Table 1, the total number of the enrolled participants in this study was ninety-six (96) and, sixty-four (64) were mothers/caregivers who have or had children with severe wasting and were interviewed through FGDs and 32 were interviewed through KIIs. Most of the participants were females and accounted for 80 percent (n = 77). The age of the participants ranged from 19 to 68 years. Most of the participants, namely fifty-eight (58) had completed primary education and most of them engaged in farming (n = 49).

**Awareness of malnutrition among study participants.** *Meaning of malnutrition.* Participants had different perspectives of what malnutrition meant. To some, the meaning of malnutrition was centred on child appearance and the implications or consequences they usually observed in children with signs of malnutrition. Majority of participants understood malnutrition as symptoms accompanied by fever and bodily weaknesses.

"...malnutrition is the food deficiency in the body, a child lacks enough nutrients, a child gets regular food but not enough and when that persists is when malnutrition develops" (IDI1_Parent_Kyela DC)

*Symptoms and signs of severe wasting.* Across the study regions, participants were able to mention symptoms of acute and severe malnutrition. The most mentioned signs and symptoms include swollen legs, face, stomach and hands.

*"You can recognise a child with malnutrition firstly by seeing how the skin is elderly-looking with wrinkles and legs become swollen" (IDI_CHW_Chemba).*

Furthermore, CHWs mentioned more technical methods in knowing signs of malnutrition by using MUAC to measure nutritional status. This is supported by one respondent who said that:

*"...when you take a child MUAC and, it reads at the red colour... means a child is malnourished" (IDI8_CHW_Kyela_Magereza).*

*Causes of malnutrition.* The study has established that spending a significant amount of time in local alcohol centres, and hence, investing insufficient time in child-caring and lack of

**Table 1. Summary profile of study participants.**

| | | District | | | | Total |
|---|---|---|---|---|---|---|
| | | **Kyela** | **Mbeya** | **Chamwino** | **Chemba** | |
| Sex | Male | 5 | 4 | 6 | 6 | 21 |
| | Female | 19 | 17 | 18 | 21 | 75 |
| Age categories | 19–25 | 4 | 6 | 3 | 9 | 22 |
| | 26–45 | 15 | 12 | 16 | 15 | 58 |
| | 46–59 | 3 | 2 | 4 | 0 | 9 |
| | 60+ | 2 | 1 | 1 | 3 | 7 |
| Education | Not educated | 1 | 0 | 7 | 5 | 13 |
| | Primary | 16 | 17 | 11 | 14 | 58 |
| | Secondary | 6 | 1 | 4 | 4 | 15 |
| | Degree | 1 | 3 | 2 | 4 | 10 |
| Occupation | Unemployed | 2 | 1 | 1 | 4 | 8 |
| | Business | 8 | 3 | 9 | 4 | 24 |
| | Famer | 10 | 14 | 10 | 15 | 49 |
| | Pastoralist | 1 | 0 | 0 | 0 | 1 |
| | Employed | 3 | 3 | 4 | 4 | 14 |
| KII | | 8 | 8 | 8 | 8 | 32 |
| FGD | | 16 | 13 | 15 | 20 | 64 |

basic knowledge on what they should feed their children for a balanced diet despite the fact that most of these required sources are affordable and locally available in their environment.

*"The malnutrition problem in our area is much caused by the way people prepare food for their children; it is so difficult because of the time. . .. . .. Most of the people here are keeping themselves busy in their income generation activities and they lack time to deal with their children" (IDI1_Parent_Kyela DC).*

In Chamwino and Chemba districts, while they mentioned other causes such as lack of nutritious foods, and poor breastfeeding practices, study participants in those areas pinpointed the environment as the cause of malnutrition.

*"Because of the dirty environment, a small child cannot control himself/herself. . .. He/she picks and eats any dirt comes across, and he/she may even eat chicken poop. So, malnutrition is caused by the dirty environment . . ." (FGD Parents Chemba Churuku).*

*Perceived magnitude of malnutrition.* There were variations in participants' description of the perceived magnitude of the problem in all districts and by categories of study participants. Some described as 'average.'

*"It is just an average. . .because we were educated to comply with nutrition and feeding children balanced diet" FGD_Parents_Chamwino_Mvumi Mission*

And other site like Dodoma they perceived the magnitude to be higher than what has been reported because they come across many people with malnutrition.

"*According to statistics they are not many. . . but in reality, if you pass through the households, you will realise that there is a problem. The problem is there*" (IDI_15_Community_leader_Chamwino_Mvumi Mission)

*Treatment experience of a child with severe wasting*. Treatment experiences of severe wasting showed that participants from most of the sites received the Ready to Use Therapeutic Food (RUTF) from the facilities.

"*The first service that is available in this facility is counseling, and when a child is found with malnutrition, they give us the medication which is nutritional food. . .If a child is suffering from other diseases the healthcare provider's treats a child accordingly*". (IDI1_Parent_Kyela DC_Ipinda).

*Perception on the quality of services*. Most of the participants have observed and reported good treatments and care, something that made them have a positive opinion with regards to the quality of care they received when their children were sick.

"*The service provided is good because many malnourished children recovered thanks to the treatment. This shows how good the service is*" (IDI10_VEO_Mbeya Dc_Idimi).

*Feeding practices of malnourished children*. Most of caregivers understood the importance of feeding their children properly, however with very few practicing. Some of participants fed their children less frequently than it is required.

"*For the malnourished children, you can feed them all the time. When they eat, they sometimes get hungry because they are not eating well. When they continue eating, they can eat four or five times per day. . ..*" (FGD1_Parents_Kyela DC).

As revealed by CHWs, community leaders also reported that caregivers of malnourished children were oriented on feeding practices for malnourished children and this enabled children to become well, as most of them adhered to it. A religious leader of Kyela elaborated:

"*The child should drink porridge in the morning and feed him pawpaw and vegetables at 10 a.m. She can also give him the food with beans, fish, or fat, and in the afternoon and evening, she will feed him food with milk*" (IDI6_Religious leader_Kyela_Indandala)

**Community perceptions/beliefs/customs related to severe wasting.** *Community perception about malnourished children*. Study participants reported different perceptions regarding malnutrition. Some thought malnutrition was caused by breastfeeding mothers who had sexual encounters with multiple partners rather than their husbands or those with pregnancies while having young children.

"*Perhaps this woman spoiled her child. . .. . ..she has another pregnancy while this child is still young and is called in Kiswahili "kabemendwa"* (IDI16_ Hcp_ Mbeya. Furthermore, DC_Inyala)

Some participants reported to associate malnutrition with witchcrafts as the following quote illustrates:

"A child is bewitched, most of the times you hear may be the grandmother is the one who did this, sometimes mother-in-law became angry because I did this, they have bad beliefs that they are bewitched" (IDI_ 2_Hcp_Kyela_Indandala)

A religious leader in Mbeya confirmed the widely-shared beliefs in witchcraft that shapes community and caregiver actions.

*"A large percentage of people who are not educated they believe that they are bewitched, and they eventually go to traditional healers and those traditional healers smear their whole bodies with the traditional medicines. . ." (IDI13_Religious Leader_Mbeya Dc_Idimi)*

Stigmatisation of a malnourished child was not observed in the study settings, however, a mother or caregiver was the one who was stigmatised and blamed for having a malnourished child and hence opted to send their children to their parents. Furthermore, participants elaborated that the tendency of sending a child to grandparents was another reason that increased the problem of malnutrition.

*". . .still they have negative perceptions because, for instance a woman who gets unspaced pregnancy does not take care of her child and sometimes send her to grandparents, thus the child is likely to become malnourished" (IDI_CHW_Chamwino_Ikulu).*

In some instances, caregivers believed that if a child was born by a thin mother, there was no possibility that the child would be fat. Such beliefs may contribute to ignoring any signs of weight loss among caregivers/mothers.

*"To add on that, most of times during discussions there were tendencies of saying that how will one be fat while their mother isn't. . . . . .They were saying I am thin. . . how will my son be fat?" (FGD_Parents_Chamwino_Chamwino Ikulu).*

On the other hand, the religious leaders interviewed in the study believed that malnutrition is accompanied by a curse. In line with this statement, a quote from one of the religious leaders said that:

*"There are some people who most of the time hide their malnourished children inside their houses believing that it is a curse, or that others will get infected if they laugh at them, so those are beliefs that exist in our society" (IDI15_ngo_Inyala_Mbeya dc).*

*Role of community health workers (CHW) as perceived by community members.* Community health workers have a central role in providing education, counseling and support for malnourished children.

*"They told me to add effort on feeding the baby, she said at seven, nine and eleven we should feed her porridge. . ..And if we eat fish I can boil it for him and feed him" (FGD_parents_Chamwino_Chamwino Ikulu).*

CHWs were also reported to provide home-based/outreach/house-to-house care to identify children with malnutrition and to make subsequent follow ups for malnourished children. For example, in Kyela, CHWs provide nutrition education to parents of the under-five children.

*"They go to the community house to house and provide nutrition education to mothers".*
*(IDI6_Religious leader_Kyela_Indandala)*

In addition, CHWs participated in providing some services at the health facilities (e.g. weighing children and, encouraged mothers/caregivers to visit health facility).

*"I was happy because after starting to attend health facility and being diagnosed I was given Plumpy'Nuts for a week. I adhered to what I was counseled that a baby should eat four and a half sachets of Plumpy'Nuts which I did the same. When I went back after a week his weight increased by one kilogram and MUAC measured was good. Then, addition of Plumpy'Nuts for seven days was offered. When I went back again, I was told the Plumpy'Nuts I was given last time were enough so they wanted to know the progress of the child only. Therefore, after that service the child was doing well, and I found it good because the service provider counsels me with love and for the benefit of my child. . .." (FGD_parents_Chamwino_Chamwino Ikulu).*

**Caregiver and community response on malnourished child.** *The first place they go for malnutrition treatment.* In our study settings, it was revealed that most of the participants were taking their children to healthcare facilities in comparison to any other responses with regards to malnutrition. However, participants mentioned some other places that other parents were taking their children in seeking for malnutrition treatments. Those other mentioned places include traditional healers and pharmacies.

*"At the hospitals, they are doing diagnosis of diseases, that's why they rush there and when they see their children are not improving that is when superstitious beliefs come in and they start advising each other to go to traditional healers because of the possibility that a child is bewitched" (IDI_Parent_Kyela DC).*

*Health seeking patterns/pathways.* Health-seeking pathways for a child with signs of malnutrition reveal a pattern in which caregivers would often seek help either from a nearby medical laboratory or health facility. For instance, in Chamwino, caregivers reported that mothers would first go to the nearest laboratory and then proceed to health facility. Alternative medicine options were also mentioned.

*"There are two places. . . there are those who educate themselves if I find my child not well the first place to visit should be the closest facility" (FGD_parents_Chamwino_Chamwino Ikulu).*

It appears mothers/caregivers took their children to hospitals when conditions were deteriorating and after frequent illness as indicated below:

*"When you see your child in such state. . ... you take him/her to the hospital for more examination about your child" (FGD_parents_Chamwino_Mvumi mission).*

*Reason for preferring health facilities.* Caregivers gave different reasons for sending their children to health facility. It appears that caregivers have faith in services offered at health facilities. One participant also urged caregivers to be inquisitive when they visit facility in order to understand their child's health conditions.

*"People go for service. They know if they visit the facility, they will get the service and if you meet the doctor, he will ask for laboratory checkups and results will be shared. . .. The problem is that the doctors are not explaining the results to us because we cannot read. . .. You give him the laboratory results and he prescribes drugs to be taken home without explaining the problem of the child" (FGD_parents_Chamwino_Chamwino Ikulu).*

Those parents who reported taking their children to traditional healers revealed that they did it out of their shared ethnic beliefs which have been passed over generations.

*". . . . .someone brought the child and she was told to go to the ward, the baby was out of water, she escaped and she is my clan relative. She was asked why she came back, she said she wanted to take the baby to the traditional healer . . . So, it is more of belief, one believes this is right and this is not" (FGD_parents_Chamwino_Chamwino Ikulu).*

**Identified reasons for negative/positive practices.** *Motivations for early healthcare seeking.* The main reason explained by most of the participants for early healthcare-seeking is attributable to knowledge on the symptoms of malnutrition. Most of the participants argued that, in many cases those parents or caregivers who took their children to early care/treatment were aware of the symptoms of malnutrition. Apart from that, being with age was also discovered to affect parents'/caregivers' decisions to take their malnourished children early to the treatment. To illustrate, young mothers didn't have enough confidence of taking their children for medical treatments compared to adult or experienced mothers who sent their children early to hospital.

*"What motivated me to complete all visits was the fact that a child has already initiated nutritional therapy, if I stopped going in those other visits it would mean that the child condition would go back to that deteriorated state. So, I decided to persevere with all visits nurses planned in order to get my child health back" (IDI1_Parent_Kyela DC).*

Strengthening village nutrition days was reported to promote health-seeking among caregivers since it provided health education and services in hard-to-reach areas.

*"Also, the presence of CHWs who are going to check these children often and they give education and there's a special nutritional day that I attend as well and, in these days, women are just given enough education and that's why you find it is impossible most of the time to get a child who has malnutrition in my ward" (IDI7_Govt. leader_Kyela).*

Some study participants mentioned that the decisions to take their malnourished children to the health facilities were motivated by their husbands, mothers and their sisters-in-law.

*"I think the person who motivates more is your husband. This is because other people like a neighbours may not know if your child is sick, that means you, as a mother and your husband are the one who know the real condition of your child every time" (FGD_parents_Chemba_Churuku).*

Some of the study participants recommended that RUFT be available in large quantities at the health facilities.

*"What motivated me to complete all visits was the fact that a child has already initiated nutritional therapy, if I stopped going in those other visits it would mean that a child condition would go back to that deteriorated state, so I decided to persevere with all visits nurses planned in order to get a child health back" (IDI1_parent_Kyela DC).*

*Barriers for early healthcare seeking among community members.* Most of the parents or caregivers have reported that reasons for late healthcare seeking for their malnourished children were due to lack of knowledge regarding the signs and symptoms of malnutrition, nature of the economic activities, poverty, negligence among parents, as well as male dominance.

*"As I said before, it is poverty. Most of them when they realise the problem, they think it may be expensive, so they don't go to hospital. Instead, they use traditional treatment to avoid costs. It is only that" (IDI_15_Community_leader_Chamwino_Mvumi Mission).*

Distance to healthcare facility was another reason for delays in healthcare seeking among some parents. The participants added that some of caregivers are staying far from health facilities, and therefore, they find it difficult to attend health facility due to transportation cost.

*"The hospital is distant and economic situation is very bad, so a person cannot afford the transportation fee as well as contribution fee to the hospital" (IDI9_CHW_Mbeya DC_Idimi).*

Another barrier mentioned was the seasonal factor, for instance, majority of people in Dodoma and Mbeya depend on agriculture for their living. During the farming season, even if the parent has a sick child, bringing the child to hospital is not a priority as they prioritise the farming activities, leading to neglect.

*"Yes, especially during the cultivating season as you said. In this society, everything 'stops during cultivating season. All the community focuses on cultivating even if they have a sick child . . ." (IDI_15_Community_leader_Chamwino_Mvumi Mission).*

Participants explained that caregivers of malnourished children sometimes became demoralised by healthcare providers who talked with them abusively particularly when caregivers delayed attending health facility.

"Fear to be harassed by nurses in clinic when delaying bringing a malnourished child to health facility" (IDI9_CHW_Mbeya DC_Idimi)

**Reasons for defaulting.** The following were reasons mentioned for parents/caregivers to default malnutrition treatment and these include interruptions of economic activities, negligence/lack of knowledge, slow recovery, location of the facility and beliefs in witchcraft. All participants who were interviewed shared the same opinion. The following quote illustrates:

*"Because when one is at the hospital, it means all other activities have stopped. So, when they stay there and think about, let say, if one had a farm then they are no longer going for cultivating. Therefore, that makes one to leave the hospital and interrupt treatment for the sake of other services" (IDI_15_Community_leader_Chamwino_Mvumi Mission).*

*"You might find one is not close to the facility, she has no fare, but if the baby is doing well and one has no fare for transport why should she visit the facility while she doesn't even have transport fare" (IDI_Religious Leader_Chamwino_Chamwino Ikulu.)*

Further, participants reported lack of support from their partners/spouses when a child is suffering from malnutrition. Lack of support is mentioned as one of the major culprits to defaulting malnutrition treatment at health facility. Similarly, they further lamented that they were blamed by their spouses for making the child sick.

*"You find you get a lot of challenges. You tell your spouse to come and do this . . . He tells you to deal with your situation. . . .You tell him to buy even powdered milk. . . .He tells you have destroyed the child..?" (FGD_Parents_Chemba_Chambalo).*

Furthermore, CHWs also reported lack of education among caregivers and highlighted the importance of completing treatment for a malnourished child. Participants explained that they sometimes didn't complete treatment due to lack of education; if a malnourished child did not show good progress during treatment, most of the caregivers stopped the treatment and decided to go for another alternative.

*"I think it is lack of education and others ignore to come back to clinic to follow up treatment because they say their children do not recover despite attending hospital at the first time" (IDI8_CHW_Kyela_Magereza).*

Unavailability of some medicines including therapeutic foods, for example RUTF, discourages mothers to continue with treatment for their children. Another reason for defaulting reported by some of respondents is abusive language used by few healthcare providers. Respondents mentioned that such hostility may be due to deteriorated conditions of their children or their loss to the appointment/follow up visits for treatment; this made mothers to default from the service

*". . .some mothers when they see their children who are malnourished still losing weight, they fear to attend health facility because some nurses can abuse them and judge them as lazy mothers" (IDI_CHW_Chamwino Ikulu).*

Shortage of healthcare providers was also mentioned as one of the causes for caregiver to stop treatment; inadequate number of health staff available at hospitals can make caregivers spend a lot of time waiting for treatment for their children.

*". . .. The providers should be increased. We come and stay here a long time and the children cry a lot" (FGD_Parents_Chemba_Chambalo)*

## Discussion

The study reported the pathway on healthcare seeking behaviour among mothers/caregivers of children with severe wasting in Tanzania. The findings have addressed the gap on reasons why mothers appeared late to the facility and sometime defaulted during the treatment.

The Government of Tanzania is implementing different interventions on creating and increasing community awareness on the importance of early seeking of the health services

[23]. Most of the participants in our study took their malnourished children first to healthcare facilities as compared to previous findings that identified traditional healers and pharmacies as the first point of treatment for severe wasting [24, 25].

However, some participants associated malnutrition with evil spirit or witchcraft; they believed that a child could be witched by the neighbours or enemies, and therefore, opted to attend to traditional healers. These were mothers/caregivers who attended to traditional healers after they observed that hospital treatment did not bring expected outcomes as quick as they would wish. The findings were similar to Hooft et al. [26] as most mothers/caregivers used both biomedical and traditional services simultaneously to treat their children during illness cycle. This can be due to the nature of our family health-seeking behaviour being influenced by people around them.

In general, the study revealed health-seeking pathways for a child with signs of malnutrition in a mixed pattern that caregivers would often seek help either from a nearby medical laboratory or pharmacy. Then, if there was no improvement in child's conditions, they went to health facility. Alternative medicine/traditional healer option was also sought if treatment progress was poor at the facility. Moreover, few participants reported to go first to traditional healers and then health facility if no improvement. The findings were similar to a study in Ethiopia [15] where mothers/caregivers took their children to traditional healers for treatment in initial stage. Families treated their children themselves and if the condition did not improve, they would visit traditional healers who would provide medication in the form of herbs and massages. They ignored medical treatment, thus delaying going to health facilities and jeopardising the efforts of the government and other stakeholders in promoting early health-seeking behaviour.

Reasons that motivated mothers/caregivers to send their children with severe wasting early to the health facility were almost similar across all regions of our study. Most of the participants argued that those parents/caregivers who took their children to health facilities early had knowledge and were aware of the malnutrition including symptoms and signs. The findings were also similar to other studies conducted elsewhere [27, 28] where decisions on early severe wasting treatment or other illnesses were influenced by low levels of awareness and lack of knowledge on acute malnutrition/other illness in children under-five. However, this is also akin to the study conducted by Lima et al. [29] which revealed that one factor motivating/discouraging mothers to go early for treatment was their understanding on the risk of the disease. This can be due to limited information regarding childhood illness including severe wasting and scaling up of the severe wasting service.

In addition, we found lack of education among caregivers contributed to incomplete treatment for a malnourished child. Also, other caregivers stopped the treatment and opted for alternative treatment when a malnourished child did not show good progress early during treatment. Similarly, other studies also reported that decisions on the treatment of severe wasting were influenced by their low levels of awareness and lack of knowledge on malnutrition in children under-five [11, 12, 28].

In other cases, mothers/caregivers took their children to traditional healers due to their beliefs on witchcraft. However, it was interesting to find that some of the interviewed traditional healers advised mothers/caregivers to take their children to hospital upon realizing the baby's condition required medical attention. Similarly, other study conducted in Uganda [26] reported reasons for opting traditional healers and the reasons include convenience services, peer influences and beliefs in traditional causes of illness. In addition, traditional healers were regarded as traditional members of the community who had native knowledge, and their treatment was perceived to be more affordable than other services [15]. Based on those mother's/caregiver's experiences, there is a need to orient tradition healers on early signs and symptoms

of severe wasting so that they can make timely and appropriate referral once they receive a child with malnutrition.

Most mothers were occupied with farming activities whereby during farming seasons, even if the parent had a sick child, the priority was not to take that sick child for treatment, but rather to go for farm activities; they neglected the child. Likewise, a study in Ethiopia showed similar findings, which identified workload as a barrier to health-seeking among mothers. Mothers explained that they were occupied with a lot of household activities such as cooking food and taking care of other children and their husbands; hence bringing a child for health care was compromised with household tasks [28].

Poverty was another barrier of health-seeking behaviour since when medicines were out of stock at health facility caregivers should purchase the medicines for their children. Moreover, when health facility was far away, caregivers needed to pay for transportation and food. Similar findings were reported in studies in which mothers were concerned about the distance, geographical location and lack of money for transportation during health-seeking for their malnourished children [28, 30]. This was further supported by a study from Uganda that explored factors influencing compliance and health-seeking behaviour for hypertension, which demonstrated that patients with a better socioeconomic status had good adherence to health seeking behavior [31].

Gender issues are demonstrated in our findings where male dominance was a barrier to health-seeking behaviour among mothers. Caregivers had to seek advice from their partners before taking a child for treatment. Some communities of Tanzania are headed by men and men are the ones who have discretionary power over decision making, including decisions on health-seeking when a child is sick. This was also similarly documented by Chakrabarti et al. [8] and, Muriithi [32] who reported the existence of cultural beliefs and male-dominance in their communities, which were associated with the well-being of the male child. The final words of the head of the family were things which affected mothers' behaviour of seeking curative health care services. In addition, Abubakar et al. [21] reported that fathers were the final decision makers concerning where the child would be taken for treatment. Therefore, as a strategy to address the issue, other studies recommended involving men in the treatment of children and implementing gender-sensitive interventions [8].

Curiosity of mothers/caregivers about their children's health conditions was the main reason for sending children to a health facility; in health facility, there is assurance of getting medical check-ups, treatment and referral to other higher-level facilities. Similar findings were reported by a study on factors related to health-seeking behaviour, which revealed that childhood illness, education level, marital status and geographical location were associated with health seeking behaviour [5].

Some mothers/caregivers opted to go to traditional healers as the first response. These mothers/caregivers attended to traditional healers after they observed that hospital treatment did not bring expected outcomes as quickly as they would wish. The findings were similar to Hooft et al. [26] as most mothers/caregivers used both biomedical and traditional services simultaneously to treat their children during illness cycle. This could be due to the nature of the family health-seeking behaviour being influenced by people around them.

Proximity of health facilities facilitates early healthcare seeking behaviours among parents/caregivers since having access to a health facility means mothers/caregivers do not spend a lot of time walking or a lot of money for transportation when seeking for treatment. The findings were similar to a study on health seeking for children with diarrhoea [27]; the participants were motivated by availability of nearest health facilities within 15 minutes walking distance. This can be explained by the fact that accessibility to heath facility saves time and money for patients.

A health and nutrition day promoted health seeking among caregivers. This is the platform where parents/caregivers with children under-five years old meet for nutritious activities such as screening for malnutrition in children under-five years, nutrition education and counseling to mothers on appropriate caring of their children in addition to feeding children nutritious diets. Sometimes, guidance and advice on the preparation of food for children is also given to mothers/caregivers. Likewise, Kabita, Rupali, and Moni [33] informed that in order to improve knowledge, awareness, health behaviour on myths and misconceptions on health and nutrition issues, there is a need of engaging the beneficiaries directly through community mobilization like village heath gatherings.

The availability of supplies at health facility, for example RUTF, motivates mothers to go to facility early when their children are identified with severe wasting [10, 13, 19, 30]. It was reported that poor quality of health services including continued absence of essential drugs deterred people from attending healthcare facilities when they were sick or disappointed patients [10, 13, 19, 30, 34]. Previous studies have shown that lack of supplies is one of the barriers of early health-seeking as it delays severe wasting treatment among under-five children suffering from acute malnutrition; the supplies identified as lacking mainly include F-75, F-100, ReSoMal, Plumpy'Nut, and Antibiotics [10, 13, 19, 30, 34].

Harshness and unfriendly attitude of healthcare providers during the treatment affected health seeking care to mothers. Mothers/caregivers when they took their children who were seriously ill to the facility, healthcare providers become angry and abuse them in front of other clients. The situation embarrasses most of the mothers. The same was reported by Mannava et al. [35] and Pravana et al. [1]. Healthcare providers' attitudes and behaviours affect patient well-being, satisfaction with care and care-seeking. This finding is surprising and contrary to popular opinions given negative attitudes towards patients from care providers usually discourage them from adopting proper and positive healthcare-seeking behaviour. The finding is not in agreement with previous research reports [1, 5, 6, 36].

## Study limitations

Our findings should be interpreted in light of some limitations. We acknowledge that some participants might have modified some of their responses to our questions upon realising that the study team had clinical/nutritional backgrounds. This might have led to underreporting of some of the cultural norms and other related views on malnutrition. Nevertheless, this highlights the need for ongoing engagement/communication initiatives at the community level to sensitise communities on childhood malnutrition.

This study was conducted in two regions of Tanzania. While the results are useful in informing future implementation of nutrition interventions, the findings may not be necessarily applicable to other contexts. Nevertheless, the findings might be applicable to settings with similar social and economic backgrounds, but caution need to be exercised in transferring the findings of this work to other settings.

## Conclusions

The results reaffirm how a programme on integrated management of severe wasting in Tanzania should encompass sociocultural factors that negatively influence mothers or caregivers of children with acute malnutrition. The programme should focus on engaging community structures including traditional healers, religious and village leaders to address prevailing local beliefs and sociocultural factors. We recommend, efforts to improve health-seeking for malnutrition among caregivers/mothers should take a socio-ecological perspective and as

demonstrated by health-seeking patterns for malnutrition, it is important to address barriers and enablers in the individual, community, and policy manner in a holistic manner.

## Supporting information

**S1 Appendix. FGD interview guide.**
(DOCX)

**S2 Appendix. KII GUIDE.**
(DOCX)

**S3 Appendix. Codebook.**
(DOCX)

**S1 Checklist. STROBE checklist.**
(DOCX)

## Author Contributions

**Conceptualization:** Elizabeth J. Lyimo, Maria Msangi, Rose V. Msaki, Magreth Rwenyagira, Ramadhan Mwiru, Germana H. Leyna, Ray M. Masumo.

**Data curation:** Aika Lekey, Mangi Ezekiel.

**Formal analysis:** Mangi Ezekiel, Ray M. Masumo.

**Funding acquisition:** Ramadhan Mwiru, Patrick Codjia, Germana H. Leyna.

**Investigation:** Elizabeth J. Lyimo, Anna J. Zangira, Rose V. Msaki, Aika Lekey, Magreth Rwenyagira, Ramadhan Mwiru, Mangi Ezekiel, Germana H. Leyna, Ray M. Masumo.

**Methodology:** Elizabeth J. Lyimo, Maria Msangi, Rose V. Msaki, Magreth Rwenyagira, Ramadhan Mwiru, Mangi Ezekiel, Ray M. Masumo.

**Project administration:** Ramadhan Mwiru, Patrick Codjia, Germana H. Leyna.

**Software:** Mangi Ezekiel.

**Supervision:** Patrick Codjia, Germana H. Leyna, Ray M. Masumo.

**Validation:** Patrick Codjia, Germana H. Leyna, Ray M. Masumo.

**Writing – original draft:** Elizabeth J. Lyimo, Maria Msangi, Anna J. Zangira, Rose V. Msaki, Aika Lekey, Magreth Rwenyagira, Ramadhan Mwiru, Patrick Codjia, Mangi Ezekiel, Germana H. Leyna, Ray M. Masumo.

**Writing – review & editing:** Elizabeth J. Lyimo, Anna J. Zangira, Rose V. Msaki, Aika Lekey, Patrick Codjia, Mangi Ezekiel, Germana H. Leyna, Ray M. Masumo.

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
