## [Decision Letter · Decision Letter 0]

26 Sep 2023

PGPH-D-23-00757

Mother’s healthcare-seeking behaviours in Dodoma and Mbeya regions of Tanzania- A qualitative study of children with acute malnutrition

Dear Dr. Masumo,

Thank you for submitting your manuscript to PLOS Global Public Health. After careful consideration, we feel that it has merit but does not fully meet PLOS Global Public Health’s publication criteria as it currently stands. Therefore, we invite you to submit a revised version of the manuscript that addresses the points raised during the review process. Our reviewers have raised very important concerns which ought to be fully addressed. Please, make sure any changes to the draft are highlighted to facilitate the re-assessment and the progress of your manuscript. 

We look forward to receiving your revised manuscript.

Kind regards,

Razak M Gyasi, PhD, PD

Academic Editor

Journal Requirements:

Additional Editor Comments (if provided):

Reviewers' comments:

Reviewer's Responses to Questions

**Comments to the Author**

1. Does this manuscript meet PLOS Global Public Health’s publication criteria? Is the manuscript technically sound, and do the data support the conclusions? The manuscript must describe methodologically and ethically rigorous research with conclusions that are appropriately drawn based on the data presented.

Reviewer #1: Yes

Reviewer #2: Partly

2. Has the statistical analysis been performed appropriately and rigorously?

Reviewer #1: Yes

Reviewer #2: Yes

3. Have the authors made all data underlying the findings in their manuscript fully available (please refer to the Data Availability Statement at the start of the manuscript PDF file)?

Reviewer #1: Yes

Reviewer #2: Yes

4. Is the manuscript presented in an intelligible fashion and written in standard English?

Reviewer #1: Yes

Reviewer #2: Yes

5. Review Comments to the Author

Reviewer #1: Firstly, I would to appreciate the researchers for raising cross cutting nutrition concerns that will improve or change the current trend of treatment and prevention of severe acute malnutrition.

Secondly, I would to raise to questions that need to be addressed

1. Researches used the term acute malnutrition and severe acute malnutrition interchangeable including the research title which is not appropriate. I suggest contextualization of the idea including the research tile “Mother’s healthcare-seeking behaviours in Dodoma and Mbeya regions of Tanzania- A qualitative study of children with severe acute malnutrition”.

2. How did you come-up with 96 study participants? Which formula did you used? Justify?

Reviewer #2: The authors of the study have chosen an interesting topic to conduct a qualitative study. However there are few issues which needs to be addressed.

Specific comments to the author

1) Title: The title of the study is confusing. The words " Children with acute malnutrition" following qualitative study is misleading.

2)Abstract:

2a.In methodology part of abstract, thematic analysis which was mentioned in the main body of manuscript have to be included along with content analysis.

2b. Distance of health facility would be rather considered a health system factor than environmental factor.

3) Introduction: Good

4) Methods

4a: A qualitative study employs various research methods which includes Ethnography, phenomenology, Grounded theory and case studies. The authors have to mention which research method was done rather than using cross- sectional study.

4b: With respect to study setting, authors can simply state the regions and districts were selected based on case load using purposive sampling.

5) Results

5a: Results section also to mention regarding various themes generated from the transcripts.

6) Conclusion

6a: Can be limited to one paragraph

6. PLOS authors have the option to publish the peer review history of their article (what does this mean?). If published, this will include your full peer review and any attached files.

**Do you want your identity to be public for this peer review?** For information about this choice, including consent withdrawal, please see our Privacy Policy.

Reviewer #1: **Yes: **Abdirahman Ahmed

Reviewer #2: No

---

## [Decision Letter · Decision Letter 1]

9 Nov 2023

PGPH-D-23-00757R1

Healthcare-seeking behaviours among mother’s having under-five children with severe wasting in Dodoma and Mbeya regions of Tanzania- A qualitative study

Dear Dr. Masumo,

Thank you for submitting your manuscript to PLOS Global Public Health. After careful consideration, we feel that it has merit but does not fully meet PLOS Global Public Health’s publication criteria as it currently stands. Therefore, we invite you to submit a revised version of the manuscript that addresses the points raised during the review process.

Please, the reviewer has a few important issues to be taken into consideration before we can consider your manuscript. Critically work on the language structure and grammar in the entire draft and highlight any changes to the draft for further assessment. 

We look forward to receiving your revised manuscript.

Kind regards,

Razak M Gyasi, PhD, PD

Academic Editor

Journal Requirements:

1.Please review your reference list to ensure that it is complete and correct. If you have cited papers that have been retracted, please include the rationale for doing so in the manuscript text, or remove these references and replace them with relevant current references. Any changes to the reference list should be mentioned in the rebuttal letter that accompanies your revised manuscript. If you need to cite a retracted article, indicate the article’s retracted status in the References list and also include a citation and full reference for the retraction notice.

2. We have noticed that you have uploaded Supporting Information files, but you have not included a list of legends. Please add a full list of legends for your Supporting Information files after the references list.

Additional Editor Comments (if provided):

Reviewers' comments:

Reviewer's Responses to Questions

**Comments to the Author**

1. If the authors have adequately addressed your comments raised in a previous round of review and you feel that this manuscript is now acceptable for publication, you may indicate that here to bypass the “Comments to the Author” section, enter your conflict of interest statement in the “Confidential to Editor” section, and submit your "Accept" recommendation.

Reviewer #1: All comments have been addressed

Reviewer #2: (No Response)

2. Does this manuscript meet PLOS Global Public Health’s publication criteria? Is the manuscript technically sound, and do the data support the conclusions? The manuscript must describe methodologically and ethically rigorous research with conclusions that are appropriately drawn based on the data presented.

Reviewer #1: Yes

Reviewer #2: Yes

3. Has the statistical analysis been performed appropriately and rigorously?

Reviewer #1: Yes

Reviewer #2: Yes

4. Have the authors made all data underlying the findings in their manuscript fully available (please refer to the Data Availability Statement at the start of the manuscript PDF file)?

Reviewer #1: Yes

Reviewer #2: Yes

5. Is the manuscript presented in an intelligible fashion and written in standard English?

Reviewer #1: Yes

Reviewer #2: Yes

6. Review Comments to the Author

Reviewer #1: Manuscript is acceptable for publication.

In point-point respond my name is not mentioned. Try to add

Reviewer #2: I would like to thank the authors for submitting revised manuscript.

However, I have noticed that only the title has been changed.

The authors, need to provide explanation for the rest of the comments.

7. PLOS authors have the option to publish the peer review history of their article (what does this mean?). If published, this will include your full peer review and any attached files.

**Do you want your identity to be public for this peer review?** For information about this choice, including consent withdrawal, please see our Privacy Policy.

Reviewer #1: No

Reviewer #2: No

---

## [Decision Letter · Decision Letter 2]

11 Dec 2023

Healthcare-seeking behaviours among mother’s having under-five children with severe wasting in Dodoma and Mbeya regions of Tanzania- A qualitative study

PGPH-D-23-00757R2

Dear Dr. Masumo,

We are pleased to inform you that your manuscript 'Healthcare-seeking behaviours among mother’s having under-five children with severe wasting in Dodoma and Mbeya regions of Tanzania- A qualitative study' has been provisionally accepted for publication in PLOS Global Public Health.

Best regards,

Razak M Gyasi, PhD, PD

Academic Editor

Reviewer Comments (if any, and for reference):

Reviewer's Responses to Questions

**Comments to the Author**

1. If the authors have adequately addressed your comments raised in a previous round of review and you feel that this manuscript is now acceptable for publication, you may indicate that here to bypass the “Comments to the Author” section, enter your conflict of interest statement in the “Confidential to Editor” section, and submit your "Accept" recommendation.

Reviewer #1: All comments have been addressed

Reviewer #2: All comments have been addressed

2. Does this manuscript meet PLOS Global Public Health’s publication criteria? Is the manuscript technically sound, and do the data support the conclusions? The manuscript must describe methodologically and ethically rigorous research with conclusions that are appropriately drawn based on the data presented.

Reviewer #1: Yes

Reviewer #2: Yes

3. Has the statistical analysis been performed appropriately and rigorously?

Reviewer #1: Yes

Reviewer #2: Yes

4. Have the authors made all data underlying the findings in their manuscript fully available (please refer to the Data Availability Statement at the start of the manuscript PDF file)?

Reviewer #1: Yes

Reviewer #2: Yes

5. Is the manuscript presented in an intelligible fashion and written in standard English?

Reviewer #1: Yes

Reviewer #2: Yes

6. Review Comments to the Author

Reviewer #1: The most issues are addressed.

Reviewer #2: All the comments have been addressed.

7. PLOS authors have the option to publish the peer review history of their article (what does this mean?). If published, this will include your full peer review and any attached files.

**Do you want your identity to be public for this peer review?** For information about this choice, including consent withdrawal, please see our Privacy Policy.

Reviewer #1: No

Reviewer #2: No
